# An Overview of Hepatocellular Carcinoma Surveillance Focusing on Non-Cirrhotic NAFLD Patients: A Challenge for Physicians

**DOI:** 10.3390/biomedicines11020586

**Published:** 2023-02-16

**Authors:** Annalisa Cespiati, Felice Cinque, Marica Meroni, Rosa Lombardi, Paola Dongiovanni, Anna Ludovica Fracanzani

**Affiliations:** 1SC Medicina ad Indirizzo Metabolico, Fondazione IRCCS Cà Granda Ospedale Maggiore Policlinico, Via F. Sforza 35, 20122 Milan, Italy; 2Department of Pathophysiology and Transplantation, University of Milan, 20122 Milan, Italy

**Keywords:** HCC surveillance, non-cirrhosis, NAFLD, conventional imaging, AFP, new biomarkers, liquid biopsy

## Abstract

Non-alcoholic fatty liver disease (NAFLD) is the most common cause of liver disease worldwide and it ranges from simple steatosis to hepatocellular carcinoma (HCC). HCC represents the first liver tumor and the third source of cancer death. In the next few years, the prevalence of NAFLD and consequently of HCC is estimated to increase, becoming a major public health problem. The NAFLD-HCC shows several differences compared to other causes of chronic liver disease (CLD), including the higher percentage of patients that develop HCC in the absence of liver cirrhosis. In HCC surveillance, the international guidelines suggest a six months abdominal ultrasound (US), with or without alpha-fetoprotein (AFP) evaluation, in patients with cirrhosis and in a subgroup of patients with chronic hepatitis B infection. However, this screening program reveals several limitations, especially in NAFLD patients. Thus, new biomarkers and scores have been proposed to overcome the limits of HCC surveillance. In this narrative review we aimed to explore the differences in the HCC features between NAFLD and non-NAFLD patients, and those between NAFLD-HCC developed in the cirrhotic and non-cirrhotic liver. Finally, we focused on the limits of tumor surveillance in NAFLD patients, and we explored the new biomarkers for the early diagnosis of HCC.

## 1. Introduction

Non-alcoholic fatty liver disease (NAFLD) is defined by the presence of hepatic steatosis, detected by imaging or histology, after the exclusion of other causes of chronic liver disease (CLD) [1]. NAFLD is the most common cause of liver damage in Western countries and it is estimated to affect 25–30% of the population worldwide, according to regional differences and the exploited diagnostic tools [2]. Indeed, the prevalence of NAFLD ranges from 35% in both North and South America to 30% in Europe and Asia, with a lower incidence (28%) in Africa [3]. Moreover, NAFLD frequency increases to more than 50% in patients with type 2 diabetes mellitus (T2DM) and to 90% in obese ones [4]. Due to the pandemic diffusion of T2DM, obesity, and to the reduction of viral chronic hepatitis prevalence, NAFLD will become the leading cause of liver transplantation in the next few years [5].

The term NAFLD covers several clinical conditions, ranging from simple steatosis to non-alcoholic steatohepatitis (NASH), cirrhosis, and hepatocellular carcinoma (HCC) [6]. HCC is the most common liver malignancy and the fifth cause of cancer worldwide, with a higher occurrence in men than women (ratio men to women 2.8:1). A total of 906,000 new cases of HCC were diagnosed in 2020, thus emerging as the third cause of cancer deaths globally [7]. The spreading of NAFLD in the coming years makes NAFLD-driven HCC an important health concern, despite the relatively lower rates of HCC in the context of NAFLD compared to other CLDs. 

The overall risk of HCC onset in NAFLD patients is 17-fold higher compared to the general population, with a stepwise progression according to the stage of the disease (lower for simple steatosis, greater for cirrhosis) [8]. Notably, HCC may also arise in the absence of cirrhosis, although it is most commonly encountered in correlation with severe fibrosis [9]. As a result, NAFLD-triggered HCC has an annual incidence of 0.7–2.6% in patients affected by NASH-cirrhosis and of 0.1–1.3% in non-cirrhotic ones [10]. 

In a NAFLD setting, free fatty acids (FFAs) accumulation in hepatocytes leads to lipotoxicity, which is associated with intracellular organelle derangement, i.e., endoplasmic reticulum (ER) and mitochondrial abnormalities, hepatocellular injury, and cell death [11,12]. Both T2DM and obesity are associated with chronic inflammation and reactive oxygen species (ROS) over-production [13], insulin resistance (IR), hepatocytes death, and hepatic stellate cells (HSCs) activation [14]. Through the release of inflammatory cytokines, a pro-inflammatory milieu is created, thereby exacerbating oxidative stress, propagating inflammation, and promoting the transition to NASH and fibrosis [7,15]. Then, chronic inflammation fosters HCC initiation and expansion [16]. Likewise, genetic predispositions may also influence the progression of NAFLD up to HCC [17], and the rs738409 single nucleotide polymorphism (SNP) in the patatin-like phospholipase domain-containing 3 (*PNPLA3*) gene is the most studied genetic predictor of NAFLD-HCC [18], although several other genetic variants are under evaluation [10].

Clinical practice aims to detect cancer as early as possible through screening tests. Indeed, a good diagnostic tool should identify the disease in a timely manner, ameliorating the probability of effective treatments in high-risk populations. [19]. For these reasons, the international guidelines recommended HCC surveillance using abdominal ultrasounds (US), with or without alpha-fetoprotein (AFP) assessment, every six months in all patients with liver cirrhosis of any aetiology and in subgroups of patients with hepatitis B virus (HBV) chronic infection without cirrhosis [20,21]. Although chronic HBV, hepatitis C virus (HCV) infections, and alcohol misuse are well-established inducers of hepatocarcinogenesis, NAFLD represents the most significant emerging risk factor [22]. Despite this evidence, HCC surveillance in this context is underestimated, due to the elevated number of patients with NAFLD, the technical difficulties to perform US in obese subjects, and the occurrence of HCC in non-cirrhotic NAFLD patients.

Therefore, this review aims firstly to summarize the clinical and pathophysiological differences between HCC arising from different aetiologies. Secondly, we evaluate the latest evidence regarding HCC surveillance in cirrhotic and non-cirrhotic NAFLD patients, emphasizing the difficulties in the HCC screening of metabolic patients and, finally, we explore the future directions for the diagnosis and monitoring of HCC in NAFLD patients.

## 2. Research Strategy and Study Selection

We conducted a narrative review by searching peer-reviewed articles about HCC surveillance across aetiologies in both cirrhotic and non-cirrhotic NAFLD patients on the PubMed database. The search time limit was before November 2022 and only English papers were included in the research. We included experimental and observational studies, clinical trials, systematic review, meta-analyses, editorials, and commentaries reporting data on HCC screening tests in CLD and on new diagnostic tools used in NAFLD-HCC diagnosis. We excluded studies that did not meet the selection criteria, literature review, meeting abstracts, duplicate publications, and those concerning other primary liver cancers except HCC.

## 3. Differences between HCC across Aetiologies 

During the past few decades, incidence and mortality rates of HCC have fallen along with the decline of HBV and HCV infections, as well as the mitigation of aflatoxin exposure [23]. Conversely, countries that were previously considered at low risk, such as most European countries, Northern America, Australia, New Zealand, and South America, have seen a rise in HCC prevalence due to the growing burden of obesity, T2DM, and NAFLD [24]. The clinical, radiological, and molecular characteristics of HCC prompted us to focus on the differences between viral-, alcohol-, and NAFLD-induced HCC. 

### 3.1. HCC in Patients with Viral Hepatitis

HBV is a DNA virus that integrates into the host genome, causing a persistent necroinflammation that leads to oncogene activation in hepatic liver cells and resulting in HCC development [24]. The lifetime risk of developing HCC is between 10 and 25%, especially in cirrhotic patients and those who have an active HBV infection [25]. This risk is further increased in combination with male sex, older age, high HBV replicative levels, infection duration, an HBV genotype, coinfection with HCV or human immunodeficiency virus (HIV), obesity and T2DM, exposure to alcohol, and tobacco [26]. 

Conversely, HCV is an RNA virus associated with a 15-to-20-fold increased risk of HCC, especially in cirrhotic patients [27]. HCV-related carcinogenesis seems to be causally linked to HCV-reiterated liver injury, which primes fibrosis and cirrhosis, possibly involving intricate epigenetic controls and complex cellular signaling networks [28]. Several co-factors may enhance HCV carcinogenesis, such as male sex, Hispanic ethnicity, HCV genotype 3, longer duration of infection, coinfections with HBV or HIV, alcohol abuse, metabolic comorbidities, and smoking habits [27,29]. Interestingly, Hispanics have a higher prevalence of fatty liver and of the *PNPLA3* C > G variant, which may explain their increased risk of developing hepatic injuries and HCC, regardless of common metabolic risk factors. However, the mechanisms underlying this association are still under investigation [30]. Viral eradication with direct antiviral agents (DAAs) has dramatically softened the risk of liver cancer and death [31] but it does not eliminate HCC risk, especially in cirrhotic patients, who therefore still require HCC surveillance after a sustained virological response (SVR) [32]. 

### 3.2. HCC in Patients with Alcoholic and Non-Alcoholic Liver Disease

Together with infectious aetiologies, even excessive alcohol intake is a well-known risk factor for HCC, rising the susceptibility to hepatic malignancy by 46% for 50 g daily ethanol consumption and by 66% for 100 g daily ethanol consumption [33]. In addition, alcohol even in low amounts can increase the risk of developing HCC in males older than 60 years, and in those with impaired liver indices [34].

Besides promoting chronic inflammation and liver cirrhosis, alcohol leads to carcinogenesis by producing acetaldehyde and ROS, altering the immune system and changing gene expression [35]. The prevalence of alcoholic etiology of HCC is country-specific—ranging from 6% in the Middle East, up to 20% in Southern Europe, and 60% in Eastern Europe—and globally it accounts for 30% of HCC-related deaths [36]. In large European cohorts of patients with alcoholic cirrhosis and several risk factors (male gender, older age, and severity of cirrhosis), the annual incidence of HCC reached up to 2.9% [37]. Accordingly, a recent meta-analysis showed a 5- and 10-year cumulative risk of HCC in alcoholic cirrhosis of 3% and 9%, respectively [38]. Furthermore, T2DM, smoking, variceal bleeding, and liver decompensation are associated with the increased risk of HCC onset in the context of alcoholic liver disease (ALD) [38]. Alcoholic cirrhosis remains the second most common indication for liver transplantation, whereas the signs and symptoms of advanced liver disease do not seem to impact post-transplant survival [39,40]. In addition, women with alcoholic cirrhosis showed higher percentage of ascites and encephalopathy, but a lower transplant rejection [41]. Around 30% of patients affected by ALD do not regularly attend surveillance, with consequent delayed diagnosis, larger tumor size, and poorer outcomes than other etiologies. 

With an annual incidence of 0.44 per 1000 person years [2], NAFLD has a relatively low risk of HCC onset compared to the other etiologies. However, NAFLD-related HCC is becoming impressively prevalent following the growth of NAFLD. HCC incidence parallels the severity of NAFLD, reaching 5.29 per 1000 person years in NASH patients and up to 0.5% to 2.6% in patients with NASH-cirrhosis [2]. Notably, around 20–30% of NAFLD-related HCC develops in non-cirrhotic livers [42].

### 3.3. HCC Clinical Features and Survival across Different Aetiologies

Few studies have compared the clinical features and survival outcomes of HCC related to HBV and NAFLD [43,44,45,46]. NAFLD-HCC patients are older, with a higher body mass index (BMI), are more likely diabetic and, interestingly, HCC develops more frequently in non-cirrhotic liver compared to HBV subjects. As for tumor characteristics, some studies registered larger tumor size with better tumor differentiation [44] in NAFLD-HCC compared to HBV-HCC. A number of studies have shown that NAFLD-HCC is more unifocal than HBV-HCC [43,44], although Lin et al. [45] did not support this finding. Perioperative mortality and morbidity seem to be similar in the two aetiologies as well as in overall survival and recurrence-free survival [43,45], although in a recent study D’Silva et al. [46] found that NAFLD could be protective against systemic recurrence in comparison to HBV [46]. However, NAFLD-derived HCC is often diagnosed at severe stages, making its management more challenging [47].

Mounting evidence regarding the comparison between NAFLD and HCV-related HCC is accumulating over the time, with still conflicting results. NAFLD-HCC seems to be often diagnosed at more advanced stages than HCV-HCC, it usually occurs with a larger tumor size and infiltrative pattern, being frequently detected in non-cirrhotic livers and outside-specific surveillance [48,49], with lower AFP levels [49,50,51]. However, prognosis appears to be superimposable in the two groups. Indeed, an Italian study showed shorter survival in patients with NAFLD-HCC compared to HCV-HCC due to the delayed diagnosis, but this result was not confirmed by the propensity score analysis [48]. Other studies did not find any significant difference in morbidity, recurrence, and overall survival between the two groups [49,50]. On the contrary, Benhammou et al. [51] showed that after treatments, NAFLD-HCC patients had longer overall survival rates compared to patients affected by viral-associated HCC, with similar recurrence-free survival [51]. Finally, Hernandez et al. [52] analyzed a cohort of transplanted patients and found a higher proportion of HCV-HCC patients with vascular invasion and poorly differentiated HCC compared to the NASH-HCC group, with a shorter recurrence-free survival of 5 years in HCV-HCC patients [52].

Two studies compared the clinical outcomes of HCC in the setting of ALD and NAFLD [53,54]. NAFLD-HCC patients were older, presented a higher rate of metabolic comorbidities and were diagnosed less frequently on surveillance than ALD-HCC patients [53,54]. NAFLD-related cancer is also less likely to occur in cirrhotic livers and with ascites than ALD-related HCC [53]. Ahn et al. [53,54] reported a worse HCC presentation at diagnosis in NAFLD patients, with a higher prevalence of an infiltrative pattern, larger tumor size, microvascular invasion, and tumors more often exceeding the Milan criteria, while Kumar et al. [53,54] found similar liver and tumor characteristics between these two groups. Finally, survival rates were analogous in NAFLD- and ALD-driven HCC [53,54].

Conversely to the risk factors, several prevention models of liver carcinogenesis have been studied. Nutrition and physical activity are well-known factors associated with the prevention of HCC development, especially in NAFLD-HCC [55]. Several clinical trials showed that a diet enriched in vegetables, such as the Mediterranean diet, is associated with a lower hepatocarcinogenesis [56,57]. Similarly, monounsaturated fats are associated with a lower risk of HCC development, in both metabolic and viral patients [58]. On the other hand, the use of processed meat is associated with HCC development, whereas white meat and fish are inversely associated with a risk of HCC [59]. Similarly, high physical activity is associated with a reduction in liver carcinogenesis, independent of body weight [60]. The viral suppression of HBV and HCV decreases the risk of HCC onset [61]. Otherwise, the role of HCV eradication by direct antiviral agents on hepatocarcinogenesis is still debated [61,62]. 

Differences in the prevalence of HCC due to the predisposing background, risk factors, and preventing strategies across the diverse aetiologies are summarized in Table 1.

## 4. HCC Onset in Cirrhotic and Non-Cirrhotic NAFLD Patients

### 4.1. Shared Predisposing Factors Triggering NAFLD-HCC in Cirrhotic and Non-Cirrhotic Patients

As previously mentioned, the risk to develop HCC severely differs across various clinical background. In this regard, metabolic patients have a higher risk of HCC onset in the absence of cirrhosis, compared to those affected by viral hepatitis or ALD [63]. Furthermore, the progression rate of HCC in patients with NAFLD-cirrhosis varies considerably, ranging from non-progressive to rapid multifocal dissemination and end stage liver disease [8]. 

Several risk factors are associated with the occurrence of HCC, both in cirrhotic and non-cirrhotic NAFLD patients, such as male sex, age, smoking, and T2DM [64,65]. The risk of developing HCC is to- to four-fold higher in males than in females, probably because androgens and estrogens play an opposite role in carcinogenesis. The androgen/androgen receptor axis is intertwined with different pathways involved in tumor formation and progression, as Wnt/β-catenin and those in which cycle-related kinase (CCRK) and Nanog transcriptional factor are involved [66,67]. Conversely, the induction of the estrogen/estrogen receptor axis hampers the growth and spread of the malignancies through the inhibition of metastasis-associated protein 1 (MTA1) expression [68]. Nonetheless, recent evidence showed an ever-increasing prevalence of HCC in women, especially in the co-presence of metabolic morbidities including T2DM and obesity [69]. Therefore, the recent findings are still conflicting. Indeed, in 2021 Myers and colleagues reported that the incidence of NAFLD-HCC in women is higher than in men, especially when T2DM coexisted [70]. 

Cigarette smoke is a well-established risk factor for several cancers outbreak, including HCC. Indeed, the odds to develop HCC are 1.55 (95% CI: 1.46 to 1.65; *p* < 0.00001) in current smokers and 1.39 (95% CI: 1.26 to 1.52; *p* < 0.00001) in former smokers, as shown in a systematic review including 81 epidemiological studies conducted in 2017 [71]. Tobacco is associated with an increase in pro-inflammatory cytokines, such as interleukin (IL) 1 and 6, TNFα, angiogenic factors, and pro-fibrotic agents. All these insults prompt inflammation, apoptosis, and fibrosis deposition. Moreover, smoke leads to vascular constriction, endothelial dysfunction, and tissue hypoxia, thereby favoring hepatocellular injuries and HCC [72]. 

T2DM is the primary metabolic risk factor associated with HCC, especially in patients with NAFLD [73]. A preclinical model of *ApoE^−/−^* mice displayed a downregulation of the *Asxl2* gene, which is involved in glucose and lipid metabolism but also in tumor cells survival and migration [74]. According to the role of T2DM in this process, numerous studies demonstrated the beneficial effect of anti-diabetic drugs on the HCC risk. In particular, the anti-cancer properties of Metformin have been highlighted in both in vitro and in vivo animal models [75,76]. A recent large retrospective study, across 85,963 patients with NAFLD and T2DM followed for 10 years, determined that Metformin administration softened the risk of HCC by 21% [77]. The protective effect of anti-diabetic drugs is also confirmed for other classes of drugs, such as dipeptidyl peptidase 4 (DPP4) inhibitors and sodium-glucose cotransporter-2 (SGLT2) inhibitors, as demonstrated in the NASH-related HCC mouse models [78,79]. Other metabolic comorbidities such as obesity, dyslipidaemia and, hypertension also favor HCC, even though they exert a minor role compared to T2DM [80]. 

A plethora or genetic predictors have been pointed out to be implicated in the development and progression of NAFLD up to HCC. The most established genetic variants are those in *PNPLA3*, transmembrane 6 superfamily member 2 (*TM6SF2*), membrane-bound O-acyltransferase domain-containing 7 (*MBOAT7*), and glucokinase regulator (*GCKR*) genes [81,82,83,84]. On the contrary, it has been demonstrated that the splice variant rs72613567 in the 17β-hydroxysteroid dehydrogenase type 13 (*HSD17B13*) gene prevents severe fibrosis and HCC [85]. 

The dysregulation of the immune system contributes to NAFLD-related carcinogenesis. Increasing intrahepatic fat accumulation exacerbates the mitochondrial dysfunction accompanied by excessive ROS production and the reduction of intrahepatic CD4+ T lymphocytes, according to a study conducted in 2016 by Ma et al. [86]. Indeed, HCC onset and spreading is physiologically prevented by intrahepatic CD4+ T lymphocytes by fostering apoptotic processes in the hepatocytes that harbor cancer mutations [87]. Moreover, T-regulatory cells (Tregs) are increased in murine models of NASH. Tregs are enabled to release immunosuppressive cytokines, which contribute to hepatic carcinogenesis [88]. T-lymphocytes are also directly and indirectly modulated by the gut microbiota composition through microbial metabolites as FFAs [89]. Changes in gut microbiota species in patients with NAFLD-cirrhosis promotes butyrate synthesis, a short-chain fatty acid that induces hepatocyte proliferation, fibrosis, and HCC, whereby influencing immune cells [90]. It has been shown that butyrate is more significantly elevated in the serum and feces of patients with NAFLD-HCC than NAFLD-cirrhosis, probably due to its immunomodulatory properties [91]. Altogether, these observations support the hypothesis that gut microbiota dysregulation promotes the development of an immunosuppressive phenotype, which is conducive to HCC [92]. 

### 4.2. Risk Factors That Discriminate HCC Onset in Cirrhotic and Non-Cirrhotic NAFLD Patients

The development of HCC in NAFLD patients without cirrhosis represents a great challenge for surveillance programs, since several differences have emerged in the past years between its onset in association with, or not with cirrhosis. Known mechanisms behind this variety encompass the activation of inflammatory signaling pathways, the release of pro-carcinogenic products due to the exaggerated lipoperoxidation, and an increase in apoptosis as a result of hepatic steatosis and IR [93]. The progression from NAFLD to HCC in non-cirrhotic livers is severely influenced by metabolic factors and steatosis degree, thus showing a lower prevalence in mild conditions and a much higher incidence in patients affected by grade 3 steatosis [94]. 

In a mouse model fed an ALIOS diet, enriched in fat and liquid sugar mimicking “fast food” meals, inflammation, hyperglycemia, and IR were all associated with a pro-proliferative environment, which induces HCC growth regardless of hepatic fibrosis [95]. Accordingly, clinical studies have shown that a better glycemic control reduces the risk of developing HCC [77]. Indeed, in a population-based study performed on 392,800 NAFLD patients, Pinyopornpanish and collaborators demonstrated that T2DM, especially when in association with the male sex, an age greater than 65, and smoking habits, remarkably enhanced the risk of HCC among non-cirrhotic NAFLD patients [65]. Moreover, elevated alanine aminotransferase (ALT) levels and markers of liver inflammation, are independently associated with an increased HCC risk (hazard ratio (HR) 6.80, 95% CI: 3.00–15.42; *p* < 0.001) in patients with non-cirrhotic NAFLD, according to the role of a proliferative environment and inflammation on tumorigenesis [96].

Despite the large numbers of evidence regarding the impact of inherited polymorphisms on NAFLD progression to more severe diseases, including HCC [18,82], few studies have pointed to the diverse genetic background between cirrhotic and non-cirrhotic HCC. In 2017, a study on 765 Italian patients with NAFLD demonstrated that the *MBOAT7* rs641738 T allele is causally correlated with HCC onset, even in non-cirrhotic patients, which is likely due to its pro-inflammatory effect [97,98]. Similarly, rare genetic variants in telomerase reverse transcriptase (*TERT*) rs2242652 C > T, *TP53*, apolipoprotein B (*APOB*), proline/serine-rich coiled-coil protein 1 (*PSRC1*) rs599839 A > G, and neurotensin (*NTS*) rs1800832 A > G have been outlined as possible modifiers of the genetic risk of HCC, irrespective of the fibrosis severity [99,100,101,102]. In more detail, Dongiovanni and colleagues used a mendelian randomization analysis and a polygenic risk score (PRS), and established that fatty liver is the main driver of advanced NAFLD up to HCC and the impact of risk alleles in *PNPLA3* C > G, *TM6SF2* C > T, *MBOAT7* C > T, and *GCKR* C > T on liver damage is directly proportional to their effect size on hepatic fat accumulation [103]. The genetic risk is amplified by metabolic disturbances, albeit genetic factors may trigger cirrhosis and HCC even in the absence of metabolic comorbidities [104]. A schematic overview of the main genetic risk factors associated with HCC onset in NAFLD patients is indicated in Table 2. 

Considering immune dysregulation as a predisposing environment for HCC, in 2021 Eldafashi et al. [105] proposed that heritable alterations in the programmed cell death-1 (*PDCD1*) gene, encoding the programmed cell death-1 protein (PD-1), are implicated in hepatocarcinogenesis. The inhibitory receptor PD-1 is expressed on the surface of activated T lymphocytes and, when it is bound to programmed death ligand-1 and 2 (PDL-1 and PDL-2), it suppresses the activity of T lymphocytes and contributes to immunosuppression [105]. Although these authors were not able to replicate their findings in other cohorts, this study highlighted the importance of immunoregulatory genes in HCC, especially in patients without cirrhosis. Moreover, in a murine model of NAFLD-HCC it has been observed as a selective reduction of intrahepatic CD4+ T lymphocytes as a consequence of the dysregulation of lipid metabolism and of lipids released from fatty-laden hepatocytes. In keeping with this observation, CD4+ T lymphocytes depletion mediated tumor initiation, expansion, and more numerous lesions. The likely mechanism behind CD4+ T lymphocytes peculiar suppression in NAFLD seems to be due to their more pronounced mitochondrial mass compared to CD8(+) T lymphocytes, thus favoring ROS overproduction and oxidative injuries, which critically mediates CD4+ T lymphocyte death. These findings further corroborate the notion that HCC may develop rapidly, even in the absence of cirrhosis when an immunosuppressive milieu prevails [86]. In this context, immunotherapeutic approaches became appealing as the standard of care in the management of HCC [106], and the immune checkpoint blockade (ICB) has been proposed as a tool to rescue the immune control of tumors [107]. However, the clinical utility of ICB in NAFLD patients remains under definition. 

HCC onset diverges in non-cirrhotic NAFLD patients from those with cirrhosis not only for its molecular characteristics, but also for its macroscopic features. Non-cirrhotic HCCs appear to be larger at the presentation and have a higher recurrence rate, probably due to a lack of screening programs, consequently delaying the diagnosis [64,108]. Even more, non-cirrhotic HCCs were more attenuated at the computed tomography (CT) scan, due to the higher presence of necrosis [109]. A different radiological pattern was observed in the arterial and delayed phases. Fibrous septae along with fat and necrosis give non-cirrhotic HCC its mosaic appearance during the late arterial phases [110]. As a result of a capsule surrounding the lesion, contrast-CT showed a delayed washout phase in non-cirrhotic HCC compared to the cirrhotic one [110]. In addition to the differences emerged by CT, a recent study showed the differences at an ultrasound between non-cirrhotic and cirrhotic HCC. In this retrospective multicentric study, a contrast-enhanced ultrasound (CEUS) was performed in 96 patients with non-cirrhotic HCC. The lesions exhibited hyperenhancement in the arterial phase and rapid washout in the portal phase, resembling metastatic liver lesions rather than cirrhotic-HCC [111]. 

A schematic illustration of the main risk factors involved in HCC development in cirrhotic and non-cirrhotic NAFLD patients is represented in Table 3**.**

## 5. Tumor Surveillance in HCC 

As previously mentioned, HCC has a high mortality rate, with an estimated 5 year overall survival of 10–20% [112], and the frequency of liver cancer is expected to rise by more than 55% in the next twenty years [113]. Curative treatments for HCC include invasive surgical approaches, such as partial hepatic resection, image-guided tumor ablation, and liver transplantation, but not all patients with HCC are eligible candidates for these therapies. A single nodule without vascular invasion or metastasis can be surgically resected, while liver transplantation is possible only when the nodule meets the Milan criteria (a single tumor ≤ 5 cm or two to three nodules ≤ 3 cm without vascular invasion or extrahepatic metastasis) [114]. Since the early detection of HCC is crucial to reducing cancer mortality, the efforts may be imperatively focused on the discovery of novel cutting-edge strategies to screen NAFLD patients, even in the absence of a fibrous background. The current clinical guidelines suggest screenings in cirrhotic patients for HCC, by performing an abdominal US every six months, along with or without an AFP measurement [21,115]. Recently, the Japanese guidelines recommended to perform HCC surveillance every three months over the past year [116], although several studies have shown that such surveillance did not improve the detection of early HCC and did not ameliorate survival rates [117]. 

### 5.1. Abdominal Ultrasound for the Screening of HCC

Hepatologic societies worldwide largely recommended an abdominal US as the first step for HCC screening, as it is a non-invasive tool that is rapidly available and economic, and it has no radiation exposure for patients. [118]. 

At B-mode US examination, a differential diagnosis of hepatic focal lesions can be made by analyzing the tumor shape, border, margins, and intratumor and posterior echo [119]. The HCC nodular-type usually appears as a round or oval lesion, whereas HCC massive-type shows as an irregular shape [120]. HCC lesions at a US can be detected as hyperechoic, hypoechoic, or with a mosaic pattern, and the prevalence of the different echogenicity observed changes according to the tumor size. Indeed, small HCCs are often hyperechoic, because of the intralesional presence of the fatty load, which histologically correlates with well differentiated cancer areas, while as the tumor grows it becomes poorly differentiated and loses fat droplets with a consequent of US hypoechoic appearance [121,122]. When the tumor mass reached a diameter greater than 20 mm, it typically develops a mosaic pattern, which pathologically represents the characteristic growth path of the tumor [123]. Moreover, advanced HCCs also show the “halo sign” and the “lateral shadows” and both expressions of the presence of a fibrous capsule of the tumor [124,125], together with the “posterior echo enhancement”, which is a less specific sign but is detectable in almost half of the HCC nodules [126]. 

Since HCC is a highly vascular tumor, color Doppler and power Doppler can be helpful in its detection. Power Doppler examination usually shows low flows inside or around the tumor for small lesions, which reflects a feeding portal flow [127], whereas bigger nodules are often characterized by an increased flow signal with a “basket-pattern” blood flow, representing the arterial vessels that surround the tumor [128]. 

The evidence that HCC surveillance with a US is a predictor of survival comes from a single randomized clinical trial (RCT) performed in China with 18,816 HBV patients. The RCT had a low adherence (60%) but was able to show a remarkable reduction (37%) in HCC-related mortality in the monitored group [129]. Further evidence relies on retrospective observational studies showing that HCC surveillance has a survival benefit in high-risk patients from different aetiologies [130,131,132,133,134]. 

Based on the mean small HCC doubling time (usually from 70 to 120 days) [135], international guidelines recommend a 6-month surveillance interval for cirrhotic patients [21,116,136], because a longer interval between each examination results in a lower survival and poorer HCC detection [132,137,138]. Hence, it could be postulated that a shorter surveillance interval may help identify HCC at earlier stages [139], albeit in a French RCT US surveillance performed every 3 months, they detected more focal lesions but failed to diagnose HCCs < 30 mm of diameter, and demonstrated no benefits in the patients’ survival [117]. 

According to a meta-analysis, despite a relatively good sensitivity in detecting HCC at any stage (84% sensitivity, 95% CI 76–92%), a US shows only a 47% sensitivity (95% CI 33–61%) for early stage HCC [140]. 

### 5.2. Alternative Imaging Approaches for the Screening of NAFLD-HCC 

Overall, despite its many advantages, US could be ineffective in HCC screening, especially in NAFLD patients [141]. Indeed, Simmons et al. [142] identified NASH-cirrhosis, male gender, BMI, Child-Pugh B or C cirrhosis, and in-patient status as predictors of surveillance failure, with a worryingly over one-third (34.6%) of NASH-related cirrhotic patients showing inadequate US images [142]. A similar rate of severe limitations in visualization at US was detected by Huang et al. [143] in a prospective cohort of NAFLD-cirrhotic patients. Therefore, US sensitivity seems to be decreased in patients with NASH compared to other aetiologies, to the point that Samoylova et al. [144] suggested that US performed in NASH patients potentially miss 41% of HCC, with a 25% decreased sensitivity vs. other CLD aetiologies (0.59 vs. 0.84; *p* = 0.02) [144]. In line with this data, a recent retrospective study conducted in 2053 cirrhotic patients showed that among all aetiologies of CLD, NAFLD-related cirrhosis had the greater proportion of limited US visualization [145]. 

The scant US sensibility in detecting HCC in the context of NAFLD is mainly due to obesity, which is extremely prevalent in NAFLD patients [146], as subcutaneous fat accumulation may attenuate the US beam, determining poor penetration and low-quality images [147]. Accordingly, many studies reported BMI as a predictor of US failure in detecting HCC [142,143,144,145,148]. In particular, Samoylova et al. [144] showed that a US was 10% less sensitive in obese patients vs. other subjects (0.76 vs. 0.87, *p* = 0.01) [144], while Huang et al. [143] showed a five times higher risk of inadequate US images for obese patients compared to non-obese ones [143]. These results can be explained both by the thickness of the subcutaneous fat layer along the abdominal wall, which increases the distance between the liver and the transducer, and the distortion and scattering of the US beam, which is caused by fat tissue [137,138]. Furthermore, peculiar tissue properties of steatosis lead to the attenuation of the US beam with poor visualization of the entire liver [149]. Moreover, not the whole liver is easy to examine through US, in particular the subcapsular regions [150]. 

To overcome the limits of a US, especially in obese patients or in the case when the liver is not fully evaluable (for excessive intestinal gas and/or chest wall deformity), or in the setting of the liver transplantation waiting list, CT scan, or magnetic resonance imaging (MRI), which are validated alternatives in HCC surveillance, as recommended in the European guidelines for HCC [151]. Nevertheless, the same guidelines did not recommend CT or MRI as the first choice exam in HCC surveillance, due to the high rate of false-positive results and the need to use a contrast agent. Moreover, CT used ionizing radiation and the cost-effective ratio for both CT and MRI, which were not favorable for a screening program. An RCT conducted in 2013 was the first to evaluate the performance of a biannual US and an annual CT for the HCC surveillance. The authors showed that a biannual US was comparable to an annual CT for the detection of early stage HCC, with lower costs and no use of radiation, suggesting the use of biannual US as a validated tool for HCC surveillance [152].

To date, contrast-enhanced imaging methods such as CT and MRI are necessary as the subsequent steps after a US for the HCC diagnose. Indeed, the latter is primarily based on the appearance of the lesion in the arterial, portal venous, and delayed phase [153]. Typically, HCC shows a hyperenhancement in the arterial phase and a washout in the portal venous and/or delayed phases [154]. The radiological aspect of HCC reflects the vascular changes that occur during HCC development, such as sinusoid capillarization and unpaired arteries. The neoplastic cells in HCC overexpress the hypoxia-inducible factor-1a (HIF-1a), with a consequential increase in the transcriptional activity of the vascular endothelial growth factor (VEGF) and erythropoietin. These modifications lead to angiogenesis and confer the typical aspect on contrast-enhanced imaging [155]. Due to the fact that HCC diagnosis required liver biopsy only in uncertain cases, in 2018 the American College of Radiology and the American Association for the Study of Liver Disease (AASLD) introduced the Liver Imaging Reporting and Data System (Li-RADS), to standardize the terminology in liver imaging [115]. The Li-RADS system was applicable in CT and MRI and was first designed for patients with liver cirrhosis. The diagnosis of HCC was based on wash in and wash-out, the non-rim arterial phase enhancement, the size of the lesion, the presence of a capsule, and the rate of tumor growth. Ancillary features such as corona highlight, intralesional fat and/or iron sparing, mosaic and nodule-in-nodule appearance, diameter stability for two years, diffusion restriction, and mild hyperintensity in T2 could be applied to better define the lesion.

The Li-RADS score ranged from one to five, with an increasing probability of HCC: LR-1 definitely benign; LR-2 probably benign; LR-3 intermediate probability of HCC; LR-4 probably HCC; and LR-5 definitely HCC. The other three categories were explicated in the Li-RADS system: LR-NC for a non-classifiable lesion due to image degradation or omission; LR-TIV for gross vascular tumor invasion, and LR-M if the lesion was probably or definitely malignant but not HCC-specific. Typical imaging features of hepatic cysts or hemangiomas are defined by LR-1 or 2 [156].

The imaging criteria for the diagnosis of HCC have been developed in patients with liver cirrhosis, but as we showed previously, several micro and macroscopic features distinguished HCC occurrence in cirrhotic and non-cirrhotic livers. Whereas cirrhotic-HCC and non-cirrhotic HCC show several similarities at imaging, such as arterial enhancement, portal and late phase wash-out, and capsule enhancement [157], NASH-HCC less frequently shows the typical wash-out. A small study conducted on 21 NASH patients showed that 40% of NASH-HCC did not display portal or late phase wash-out at MRI [158]. Afterward, a multicentric study conducted on 107 patients who underwent surgery for suspected HCC in a non-cirrhotic liver confirmed that nearly 30% of HCC occurred in non-cirrhotic liver, which showed the wash-out phase on the MRI [159]. Comparing the imaging features of NASH-HCC and virus-induced HCC, Barat et al. [160] showed that both of them seem to have the primary criterion of Li-RADS, including portal and late-phase wash-out [160]. The presence of hepatic steatosis seems to be related to the absence of wash-out at MRI [161] but not at CT imaging [162], which possibly explains the differences of the wash-out prevalence in NASH-HCC across the studies. 

Probably due to the lack of tumor surveillance in non-cirrhotic HCC, these patients frequently showed a large solitary mass with peripheral satellites, extrahepatic spread of the disease, invasion of portal vein, and abdominal lymphadenopathy [109]. 

### 5.3. Alpha-Fetoprotein Assessment for the Screening of HCC

All the efforts of researchers should be imperatively addressed to overcome the hurdles of the early diagnosis and to improve the prognosis of HCC. Given its ever-increasing prevalence and the cancer-related mortality in the NAFLD setting, it is crucial to focus our attention on the discovery of non-invasive biomarkers with a reliable diagnostic power and clinical utility during follow-up. To date, proteins with high expressions in tumoral tissues compared to the adjacent ones are the most attractive analytes that are exploited for HCC surveillance programs. Among them, despite several limitations, AFPs still have a high *consensus* among clinicians for tumor staging, grading, and management, due to their elevated diagnostic performance for the detection of HCC [163]. According to the 2018 European Association for the Study of the Liver (EASL) guidelines on HCC, AFP was not recommended alone for the HCC screening surveillance, due to its suboptimal cost-effectiveness ratio [164]. Almost 80% of small tumors, which were less than 3 cm, had normal levels of AFP, with a low sensitivity of AFP in these lesions (sensitivity 25%) [165]. Nevertheless, a meta-analysis conducted in 2018 and including 13,367 patients showed that the sensitivity of US for the detection of small HCC in cirrhosis was significantly higher when AFP was added to the US (45% vs. 63%, respectively) [140]. According to Asian guidelines, AFPs could be included in HCC surveillance along with US [116]. The association of AFPs with US seems to be a higher benefit in viral-induced HCC rather than metabolic ones [166], but there is a lack of NASH-HCC-tailored studies. To overrule the above mentioned limits on HCC surveillance, several biomarkers were evaluated and proposed. 

The inflammatory milieu that characterizes HCC tumorigenesis involves T-regulatory lymphocytes with non-specific neutrophilia and relative lymphopenia. The consequence is an immune-mediated antitumor response, with improves HCC progression [167]. A retrospective cohort study conducted on 789 HCC patients showed that the neutrophil-lymphocyte ratio (NLR) appeared to parallelly increase with HCC severity, and an NLR greater than three was associated with a large tumor size, vascular invasion, and tumor rupture [168]. Patients with NASH-HCC seem to have a higher NLR compared to viral-ones, and NLR may be a prognostic marker for disease severity [168].

### 5.4. Novel Strategies to Non-Invasively Assess NAFLD-HCC: Proteins and Receptors

Des-gamma-carboxy prothrombin (DCP), also known as pro-thrombin induced by vitamin K absence-II (PIVKA II), is an aberrant prothrombin molecule induced by vitamin K and is overproduced during the malignant transformation of hepatocytes [169]. Therefore, DCP is a promising HCC predictive and prognostic marker, and, when it is combined with AFP, it may complement US in the early tumor detection [163]. Moreover, considering its role in the crosstalk between HCC and vascular endothelial cells, it may also be useful in identifying more aggressive tumors and advanced tumor stages, and in evaluating the prognosis after therapies [170]. Glypican-3 (GPC3) is a heparin sulfate proteoglycan that is involved in the regulation of developmental morphogenesis through the interaction with numerous growth factors [171]. GPC3 levels are elevated in HCC patients, but not in hepatitis or healthy subjects [172]. Of note, GPC3 expression levels are higher in cirrhotic livers with dysplasia compared to those without it, suggesting its potential use as a precancerous biomarker [173]. However, this biomarker has a low sensitivity and poor detection rate in blood samples compared to hepatic biopsies, but it gains sensitivity and specificity when it is assessed simultaneously with AFP [174]. Osteopontin (OPN) is an extracellular matrix multifunctional protein that is physiologically expressed in Kupffer and stellate cells, but not in hepatocytes [175]. OPN levels are positively associated with the HCC risk [176], and many studies show a better accuracy of the combination of OPN and AFP for HCC diagnosis compared to AFP alone [177]. Golgi protein-73 (GP73) is a Golgi trans-membrane glycoprotein involved in HCC cell proliferation, invasion, and migration [178]. Several pieces of evidence demonstrate its high sensitivity and specificity in HCC detection, which is even higher than AFP [179], which makes GP73 a promising HCC biomarker.

G protein-coupled receptors (GPCRs), which are known to be involved in carcinogenesis and are reported to be mutated and overexpressed in a HCC microenvironment, are currently being evaluated as potential HCC biomarkers [172]. In particular, a recent study proposed GPCRs such as the beta2-adrenergic receptor as a predictive factor for both recurrence-free and overall survival, being upregulated in HCC tumor tissues and significantly associated with poor prognosis [180].

### 5.5. Novel Strategies to Non-Invasively Assess NAFLD-HCC: Cell-Circulating Tumor DNA and Non-Coding RNAs

An innovative strategy to assess the genetic profile of primary and metastatic tumors and to dynamically track their genomic evolution is constituted by a liquid biopsy [164]. The latter may represent a good opportunity to ameliorate our ability to explore the tumor molecular signature and heterogeneity, thus favoring the discovery of feasible biomarkers into the circulation, ameliorating the tumor screening, surveillance, detection, and outcome. This approach takes advantage of the opportunity to resample body fluids over time, addressing analytes, actively or passively, that are released into the bloodstream from cancerous lesions. For instance, cell-circulating tumor DNAs (ctDNAs) are gaining an exceptional potential in the management of HCC. ctDNAs are double-stranded DNA fragments, containing genetic aberrancies that are identical to the tumor cells from which they are derived from, and are detectable in both the serum and plasma [181]. In more detail, these DNA fragments are released by apoptotic- or necrotic-injured cells and their genetic/epigenetic pattern has been associated with cancer aggressiveness in different tumors, including HCC [182,183,184]. Specifically, the assessment of the SNPs or methylation changes in this fragmented DNA may exert a central role in the determination of cancer recurrence and metastatic potential [185], to the extent that the methylation changes of the ctDNA have been linked to early tumor occurrence [186]. In a cohort of 1098 HCC patients, it has been reported that the methylation profiles of HCC tumor DNA are strongly correlated with the matched plasma ctDNA [187]. Multiple methylation aberrancies have been linked to HCC, and the hypermethylation of promoter regions has been outlined as a precocious anomaly in tumorigenic processes. Therefore, the combined assessment of ctDNAs and AFP may improve HCC detection, exceeding the previously described plasma biomarkers, in terms of higher sensitivity and better clinical correlation in discriminating HCC patients from the normal controls [188]. Plasma DNA levels have also been positively correlated with tumor size, intrahepatic spreading, and vascular invasion, being an independent risk factor for poor overall survival, recurrence, and extrahepatic metastasis [189]. Finally, ctDNA concentrations have also been associated with the response to ICB therapy with pembrolizumab, a monoclonal antibody anti PD-1, also predicting the therapeutic outcomes [190].

Furthermore, mounting evidence indicates that changes in the expression of short or long non-coding RNAs (ncRNAs) may be indicative of NAFLD worsening into HCC [191,192]. These molecules do not encode proteins, albeit they still modify the expression of target genes. Specifically, during HCC initiation, worsening, and metastatic spreading, several ncRNAs are tremendously altered, thus suggesting their potential regulatory role in these processes [192]. Thus, ncRNAs that are isolated from tumoral tissues, blood, and urine could be useful in the future as biomarkers for the early detection of HCC or to foresee the prognosis of patients, possibly bridging the gap between clinical requirements and the current needs [193].

In detail, among the short ncRNAs, we can take into account the endogenous microRNAs (miRNAs) that may dually operate as either oncogenes or onco-suppressors upon different conditions [194,195]. Their most studied aberrancies include the downregulation of miR-122, miR-15, miR-16, and miR-34a in NASH-associated HCC [196,197], and the overexpression of miR-221 and miR-101-3p [198], whereby paralleling the severity, invasiveness, and TNM classification of HCC.

Conversely, along with the short ncRNAs, also long non-coding RNAs (lncRNAs) are non-codifying molecules with transcriptional (gene activation/silencing) or post-transcriptional (mRNA splicing) regulatory properties, but with a length of more than 200 nucleotides [191,199]. In the context of NAFLD-derived HCC, lncRNA MALAT1, HULC, NEAT1, HOTAIR, and H19 have been extensively described as novel modifiers of the predisposition to liver carcinogenesis and chemoresistance [172,200].

Finally, the circular non-coding RNAs (circRNAs) are gaining overwhelming interest as biomarkers in different tumor types. They are stable scrambled exons that are resistant to endonucleases, consisting of a structure of a circular-loop RNA void of a 5-cap and 3-tail containing conserved miRNA response elements (MREs) [201]. As a consequence, circRNAs work as “miRNA sponges”, enabled to sequester more than one miRNA, smoothening their activity, and further affecting the expression of downstream mRNA through a circRNA–miRNA–mRNA pathway [202]. In patients with HCC, some circRNAs are aberrantly expressed and they may participate with pivotal events that occur during NAFLD evolution toward HCC, such as lipogenesis, fibrosis, and cell proliferation [172,203].

## 6. New Proposed Scores to Estimate the Risk of NAFLD-HCC

Apart from novel biomarkers, several scores have been proposed to predict HCC prognosis in the setting of NASH. For instance, the fibrosis-4 index (FIB-4) is a widely used score that encompasses age, platelet (PLT) levels, AST, and ALT to define the risk of advanced fibrosis in CLD. Because patients with increased FIB-4 have a higher probability of advanced fibrosis, it is conceivable that patients with high FIB-4 also have an increased risk of HCC development. A huge study conducted in 2018 on 25,947 Korean patients with a one-year follow-up showed that a FIB-4 greater than 1.45 was associated with an increased risk of HCC in NAFLD [204]. Another large retrospective European study conducted on 29,999 NAFLD patients showed that a FIB-4 greater than 1.3 was associated with HCC development within the 10-year follow-up, even without cirrhosis at the baseline [205].

Similar findings have been shown with the use of transient elastography (TE). A FibroScan through the use of a pulse-echo US acquisition (vibration-controlled transient elastography (VCTE)) simultaneously quantifies both the liver fibrosis by a liver stiffness measurement (LSM) and the liver steatosis through the use of a controlled attenuation parameter (CAP) [206]. Izumi et al. [207] performed a FibroScan to evaluate both the LSM and CAP in 1656 patients with CLD. They showed that the LSM ≥ 5.4 kPa and CAP ≤ 265 dB/m were primarily associated with the risk of HCC development in NAFLD patients. The LSM cut-off was lower when compared to viral patients (5.4 kPa for NAFLD vs. 8 kPa for HCV and 6.2 kPa for HBV). The authors also confirmed that an increased FIB-4 value (greater than 2.67) was significantly associated with the risk of HCC development in NAFLD patients [207].

To estimate the risk of advanced fibrosis and the adverse clinical outcomes among patients with NAFLD, the enhanced liver fibrosis (ELF) test has been proposed. This score considers the amount of hyaluronic acid, tissue inhibitor of matrix metalloproteinase-type 1 (TIMP-1), and the aminoterminal propeptide of type 3 procollagen (P3NP). In a large retrospective cross-sectional study, Younossi and colleagues reported that the AUROC for ELF in identifying patients with histologically or non-invasively diagnosed severe fibrosis was 0.81 (95% CI, 0.77–0.85) and 0.79 (95% CI, 0.75–0.82), respectively [208]. The simultaneous assessment of ELF and FIB-4 may have a reliable clinical utility to determine the presence of progressive fibrosis among NAFLD patients, although a recent study demonstrated that ELF and VCTE were superior to FIB-4 for all fibrosis endpoints [209], also exceeding other scores, such as the NAFLD fibrosis score (NFS) and the BARD score.

Another score that has been demonstrated to have a good accuracy in both non-cirrhotic and cirrhotic NASH-HCC was the GALAD score. The GALAD score encompasses gender, age, AFP-L3 (an isoform of AFP more specific for neoplasia), AFP, and DCP and it seems to have a high prognostic accuracy in the early detection of HCC in NASH patients (area under receiver operating characteristic curve, AUROC: 0.96). The accuracy of GALAD was high, independent of aetiologies and cirrhosis, with similar AUROCs in patients without cirrhosis (AUROC: 0.98) and those with cirrhosis (AUROC: 0.93). GALAD could detect HCC independent of cirrhosis and the HCC stage, and it became raised within a few months of HCC detection. The cut-off threshold proposed for GALAD was −1.334, and it could be used in the future in the screening program for HCC detection, especially in non-cirrhotic NASH patients [210]. Notably, the combination of GALAD and US (GALADUS score) may further ameliorate the performance of the GALAD score alone [211].

Non-invasive tests (NITs) for stratifying NAFLD patients according to the risk of developing HCC are listed in Table 4.

Finally, the role of genetic predispositions is well-known in NAFLD development and progression, and the role of the genetic risk in identifying patients at high risk to develop HCC is an area of active research. As previously described, the genetic polymorphisms in *PNPLA3* C > G, *TM6SF2* C > T, *MBOAT7* C > T, and *GCKR* C > T genes are predisposed to NAFLD progression and HCC development [212]. On the other hand, the rs72613567 *HSD17B13* TA variant seems to prevent hepatic fibrosis and HCC tumorigenesis [85]. Based on these assumptions, Bianco et al. [213] in 2021 proposed a PRS to identify the risk of HCC in NAFLD patients. A PRS based on the five aforementioned variants could be used to predict the risk of HCC in patients with NAFLD and concomitant dysmetabolism, targeting the neoplastic surveillance on metabolic patients without cirrhosis but with a high PRS [213]. Thus, a good PRS may correlate to clinically relevant genetic variants with environmental and dynamic risk factors with the purpose to acquire greater accuracy for HCC early detection [17]. A previous study performed on the general population confirmed that an increased PRS based on *PNPLA3* rs738409, *TM6SF2* rs58542926, and *HSD17B13* rs72613567 conferred a higher risk of both cirrhosis and HCC development, with a higher risk of HCC compared to the risk of cirrhosis (29-fold higher for HCC vs. 12-fold higher for cirrhosis) [18]. Moreover, in 1380 patients with NAFLD, among whom 121 had HCC, Longo et al. [82] evaluated the impact of the three variants, I148M PNPLA3, rs641738 *MBOAT7,* and E167K TM6SF2, showing that the co-presence of these three at-risk variants was related to enhanced levels of markers of liver damage, advanced steatosis, inflammation, ballooning, fibrosis, and approximately a two-fold higher risk of HCC [82]. A schematic representation of the main risk factors involved in HCC development in cirrhotic and non-cirrhotic NAFLD patients and the current and future tools for the diagnosis of HCC in non-cirrhotic NAFLD are illustrated in Figure 1.

Considering the extent of the NAFLD population, a good screening test should embrace various criteria, including high sensitivity and specificity, cost effectiveness, and availability [214]. Other predictive models have recently been proposed to stratify the risk of developing HCC and surveillance in patients. Among them, Chen et al. developed in 2022 a highly accurate diagnostic model that combines a 12-gene signature, biological pathway analysis and a machine learning algorithm, with the purpose of distinguishing between cancer and noncancerous tissues [215]. Overall, machine learning algorithms will become a powerful tool for clinicians to accurately identify patients at high risk for HCC development, and they will pave the way for the optimization of personalized therapeutic approaches [216].

## 7. Concluding Remarks

Despite NAFLD showing a low incidence rate of HCC compared to other causes of CLD, it is predicted that the rate of NAFLD-HCC will increase faster in the next few years due to its spreading worldwide.

In contrast to viral and alcohol-related HCC, NAFLD-HCC develops in non-cirrhotic livers with a higher frequency (below 20–30% of cases). NAFLD-HCC appears larger at diagnosis, with an infiltrative pattern and microvascular invasion, probably because the screening program failed more frequently in these patients.

Male sex, T2DM, age, smoking, and a higher BMI are clinical risk factors related to the development of NAFLD-HCC, also in non-cirrhotic livers. The pathophysiological mechanisms responsible for non-cirrhotic NAFLD-HCC onset include systemic inflammation, hyperglycemia, IR, and the immune dysregulation with low intrahepatic CD4+ T lymphocytes. Furthermore, genetic predisposition plays an important role in the development of non-cirrhotic NAFLD-HCC, as testified by the great accuracy of PRS to foresee HCC. Furthermore, gut dysbiosis seems to be related to non-cirrhotic HCC through the promotion of a pro-inflammatory milieu.

The international guidelines suggest performing a 6 months surveillance with a US, with or without AFP measurements, in all patients with liver cirrhosis and in a subgroup of patients with HBV chronic infection without cirrhosis. However, a US shows several limits for NAFLD patients, probably due to obesity and the steatosis-mediated attenuation of US beams. CT and MRI could be valid alternative to US for NAFLD patients, but some concerns such as the use of ionizing radiation, the costs, allergies, and chronic kidney diseases limits the use of these techniques.

AFP is often normal in patients with HCC, especially of those with metabolic origin. For this reason, AFP should only be measured together with a US examination. Several biomarkers and scores are being proposed to overcome the limits of screening surveillance and to detect early HCC, especially in non-cirrhotic patients. A liquid biopsy with the evaluation of ctDNAs, ncRNAs, and circRNAs seems to be promising, nevertheless its use is not routinary. FIB-4, LSM, and GALAD scores could help physicians better identify patients with higher probabilities of developing HCC and PRS.

The possibility of combining multiple biomarkers may offer more accurate and valuable information for HCC diagnosis than the use of a single one.

In conclusion, HCC detection in NAFLD patients is a challenge for physicians, due to the high prevalence of NAFLD, the possibility of HCC development in non-cirrhotic livers, and the limits of conventional imaging in these patients. In the context of a more personalized medicine approach, the PRS and the liquid biopsy are promising tools for the early detection of HCC in both cirrhotic and non-cirrhotic NAFLD patients.

## Figures and Tables

**Figure 1 biomedicines-11-00586-f001:**
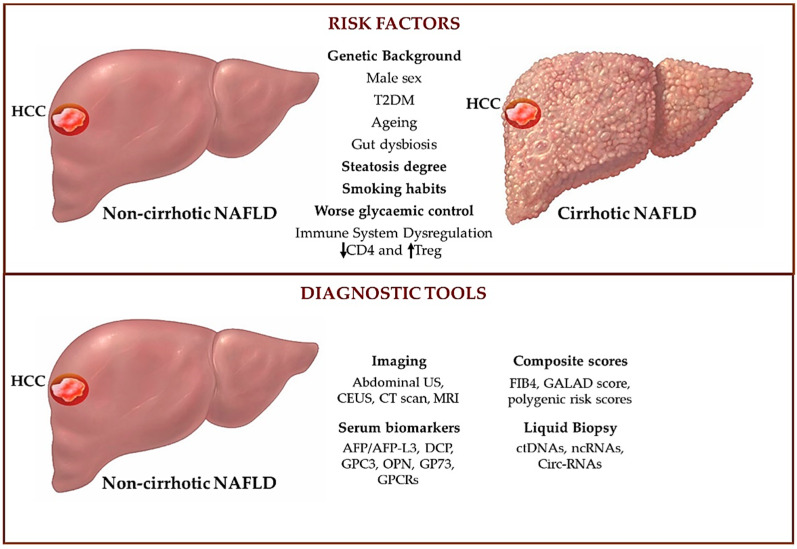
Clinical and molecular risk factors associated with the development of HCC in NAFLD patients with or without cirrhosis (**upper panel**) and current and future diagnostic tools for the diagnosis of HCC in non-cirrhotic NAFLD patients (**lower panel**). Genetic background, steatosis degree, smoking habits, and worse glycemic control are marked **in bold,** since these factors more specifically trigger HCC onset in non-cirrhotic NAFLD patients. **↓**: downregulation; **↑**: upregulation. HCC: hepatocellular carcinoma; NAFLD: non-alcoholic fatty liver disease; T2DM: type 2 diabetes mellitus; US: ultrasound, CEUS: contrast-enhanced ultrasound; CT: computed tomography; MRI: magnetic resonance imaging; AFP: alpha-fetoprotein; DCP: des-gamma carboxyprothrombin; OPN: osteopontin; GP73: Golgi protein-73; GPCRs: G protein-coupled receptors; FIB4: fibrosis-4 index; PRS: polygenic risk score; ctDNAs: cell-circulating tumor DNAs; and Circ-RNAs: circular non-coding RNAs.

**Table 1 biomedicines-11-00586-t001:** Differences in HCC prevalence according to predisposing background, risk factors, and preventing strategies across the diverse aetiologies.

	HCV	HBV	ALD	NAFLD
HCC prevalence	15–20%	10–25%	46–66%	0.5–2.6%
Predisposing background	- Cirrhosis (before/after viral eradication)	- Cirrhosis - Active replication	- Daily alcohol consumption	- Cirrhosis (20–30% in non-cirrhotic livers)
Risk factors	- Male sex - Ethnicity (Hispanic) - HCV genotype 3 - Longer duration of infection - HBV and/or HIV coinfections - Alcohol - Metabolic comorbidities - Smoking habits	- Male sex - Ageing - HBV genotype - High replicative levels - Longer duration of infection - HCV and/or HIV coinfections - Alcohol abuse - Metabolic comorbidities - Smoke habits	- Male sex - Ageing - Impaired liver enzymes	- Male sex - Ageing - Higher BMI - Diabetes
Preventing factors	- Antiviral therapy	- Antiviral therapy - Vaccination	- Alcoholic abstention	- Mediterranean diet - Physical activity

HCV: hepatitis C virus; HBV: hepatitis B virus; ALD: alcoholic liver disease; NAFLD: non-alcoholic fatty liver disease; and BMI: body mass index.

**Table 2 biomedicines-11-00586-t002:** Schematic list of the main inherited variations related to NAFLD-HCC.

Variant	Gene	Global MAF	Function	Effect	Impact	Phenotype
rs738409 C > G	*PNPLA3*	0.26 (G)	Lipid remodeling	p.I148M	Loss-of-function	↑ NAFLD, NASH, fibrosis, and HCC
rs58542926 C > T	*TM6SF2*	0.07 (T)	VLDL secretion	p.E167K	Loss-of-function	↑ NAFLD, NASH, and fibrosis
rs641738 C > T	*TMC4/MBOAT7*	0.37 (T)	Lipid remodeling	p.G17E	Loss-of-function	↑ NAFLD, NASH, and fibrosis, HCC
rs1260326 C > T	*GCKR*	0.29 (T)	Regulation of DNL	p.P446L	Loss-of-function	↑ NAFLD, NASH, and fibrosis
rs72613567 T > TA	*HSD17B13*	0.18 (TA)	Lipid remodeling	Truncated protein	Loss-of-function	↓ NASH, fibrosis, and HCC
Several	*APOB*	NA	VLDL secretion	Protein change	Loss-of-function	↑ NAFLD NASH, fibrosis, and HCC
Several	*TERT*	NA	Telomere maintenance	Protein change	Loss-of-function	↑ Fibrosis and HCC
Several	*TP53*	NA	Genomic stability maintenance	Deletion	Loss-of-function	↑ HCC
rs1800832 A > G	*NTS*	0.11 (G)	Lipid metabolism	Overexpression of circulating Pro-NTS	Gain-of-function	↑ Fibrosis and HCC
rs599839 A > G	*PSRC1*	0.24 (G)	Microtubule destabilization and spindle assembly	Overexpression of *CELSR2-PSRC1-SORT1* gene cluster	Gain-of-function	↑ HCC

MAF: minor allele frequency.

**Table 3 biomedicines-11-00586-t003:** Similarities and differences in HCC carcinogenesis between cirrhotic and non-cirrhotic NAFLD patients.

	Non-Cirrhotic vs. Cirrhotic NAFLD
Common risk factors	- Male sex - Age > 65 years - Smoking habits - T2DM - Unhealthy lifestyle - Genetic polymorphisms in *PNPLA3*, *TM6SF2*, and *MBOAT7* genes - Immune system dysregulation - Alterations in gut microbiota
Mechanisms involved in carcinogenesis	- Worst glycemic control and IR - Genetic variants in *TERT*, *TP53*, *APOB*, *PSRC1*, *NTS*, and *PDCD1* genes - Reduction of intrahepatic T CD4+ cells
Macroscopic features	- Larger lesions at the presentation - More attenuated at CT scan - Mosaic appearance in late arterial phase at CT - Delayed washout at CT - Hyperenhancement in arterial phase and rapid washout at CEUS

T2DM: type 2 diabetes mellitus; PNPLA3: patatin-like phospholipase domain-containing 3; TM6SF2: transmembrane 6 superfamily member 2; MBOAT7: membrane-bound O-acyltransferase domain-containing 7; GCKR: glucokinase regulator; HSD17B13: 17beta-hydroxysteroid dehydrogenase type 13; IR: insulin resistance; TERT: telomerase reverse transcriptase; APOB: apolipoprotein B; PSRC1: proline/serine-rich coiled-coil protein 1; NTS: neurotensin; PDCD1: programmed cell death 1; CT: computed tomography; and CEUS: contrast-enhanced ultrasound.

**Table 4 biomedicines-11-00586-t004:** Non-invasive tests (NITs) for stratifying NAFLD patients according to the risk of developing HCC.

NIT	Formula	Higher Risk
VCTE	NA	>18 KPa
MRE	NA	>3.63 KPa
BARD score	BMI ≥ 28 kg/m2 = 1 AST/ALT ratio ≥ 0.8 = 2 T2DM = 1	≥2
FIB-4 index	age (years) × AST (U/L)/[PLT (109/L) × ALT1/2 (U/L)]	>2.67
NFS	−1.675 + 0.037 × age (year) + 0.094 × BMI (kg/m2) + 1.13 × IFG/diabetes (yes = 1, no = 0) + 0.99 × AST/ALT ratio − 0.013 × PLT count (×109/L) − 0.66 × albumin (g/dL)	>0.676
ELF test	2.494 + 0.846 × ln(HA) + 0.735 × ln(P3NP) + 0.391 × ln(TIMP1)	>9.89
GALAD score	−10.08 + 1.67 × [Gender (1 for male, 0 for female)] + 0.09 × [Age] + 0.04 × [AFP-L3%] + 2.34 × log[AFP] + 1.33 × log[DCP]	≥−0.63

ALT: alanine aminotransferase; AST: aspartate aminotransferase; BMI: body mass index; HA: hyaluronic acid; IFG: impaired fasting glucose; MRE: magnetic resonance elastography PLT: platelets; P3NP: pro-peptide of type 3 procollagen; and TIMP1: tissue inhibitor of matrix metalloproteinase type 1.

## Data Availability

Not applicable.

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
