# Peer review of "An Overview of Hepatocellular Carcinoma Surveillance Focusing on Non-Cirrhotic NAFLD Patients: A Challenge for Physicians"

_biomedicines, 2023, doi:10.3390/biomedicines11020586_

Round 1

Reviewer 1 Report

In the present manuscript, Cespiati and colleagues provide a comprehensive picture of hepatocellular carcinoma occurrence in patients with NAFLD, from epidemiology to risk factors, and from potential underlying mechanisms to surveillance. Overall, the topic addressed is clinically relevant and the manuscript is well written. 

Please find below some comments to the authors:

1) The title of the review is "Hepatocellular carcinoma surveillance in non-cirrhotic NAFLD patients: a challenge for physicians." I believe that such title is too much focused as compared to the broad arguments covered by the manuscript. I suggest to amend the title accordingly.

2) Paragraps 3 and 4. The manuscrip may benefit from a Table summarizing epidemiologic and clinical characteristics of HCC occurring in patients with liver cirrhosis and those without, and according to etiology.

3) Biomarkers for HCC screening. There are several other biomarkers currently under investigation that are worth to mention (i.e. DCP/PIVKA-II, etc). I suggest to add a paragraph between 5.3 and 5.4 describing the results of the such biomarkers, in particular for those with available studies investigating the accuracy for the prediction of HCC development in patients at risk. 

Author Response

In the present manuscript, Cespiati and colleagues provide a comprehensive picture of hepatocellular carcinoma occurrence in patients with NAFLD, from epidemiology to risk factors, and from potential underlying mechanisms to surveillance. Overall, the topic addressed is clinically relevant and the manuscript is well written. Please find below some comments to the authors:

We thank the Reviewer for her/his thoughtful and thorough revision of the manuscript and for the relevant suggestions.

Please see below the point-to-point answers to the Reviewer. Changes in the manuscript are marked up using the “Track Changes” (which are in red) and highlighted in yellow.

Point 1: The title of the review is "Hepatocellular carcinoma surveillance in non-cirrhotic NAFLD patients: a challenge for physicians." I believe that such title is too much focused as compared to the broad arguments covered by the manuscript. I suggest to amend the title accordingly.

Response 1: We thank the Reviewer for the comment. We have now modified the title of the manuscript in “An overview of Hepatocellular Carcinoma Surveillance focusing in on Non-Cirrhotic NAFLD Patients: A Challenge for Physicians”, thus emphasizing the focus on non-cirrhotic NAFLD patients.

Point 2: Paragraps 3 and 4. The manuscript may benefit from a Table summarizing epidemiologic and clinical characteristics of HCC occurring in patients with liver cirrhosis and those without, and according to etiology.

Response 2: We completely agree with the Reviewer. We have now implemented the manuscript with two new tables (Table 1 and 3) to better summarize the differences between HCC development according to aetiologies and the specific features of HCC carcinogenesis in non-cirrhotic NAFLD patients.

Point 3: Biomarkers for HCC screening. There are several other biomarkers currently under investigation that are worth to mention (i.e., DCP/PIVKA-II, etc). I suggest to add a paragraph between 5.3 and 5.4 describing the results of the such biomarkers, in particular for those with available studies investigating the accuracy for the prediction of HCC development in patients at risk. 

Response 3: We really thank the Reviewer for pointing this out. We have now improved the revised version of the manuscript, by adding a chapter describing other biomarkers currently under investigation, among which DCP (5.4. Novel strategies to non-invasively assess NAFLD-HCC: proteins and receptors).

Reviewer 2 Report

The authors review the factors that affect survival in patients with HCC. Some minor issues could be improved:

1. Could the authors refer to how the Hispanic ethnic group is more potentiated in carcinogenesis by HCV?

2. Some references on HCV would be interesting to consider in this review:

DOI: 10.1016/j.cgh.2022.06.032

doi: 10.1016/S1470-2045(22)00078-X.

https://doi.org/10.1111/acer.13013

DOI : 10.6002/ect.2017.0302

doi: 10.5114/aoms.2018.80651

https://doi.org/10.3171/2022.8.FOCUS2255

3. Authors should indicate the reference number throughout the document after the citation to facilitate reading and searching. Page, 4 et al., line 4 and 6, line 31 and 34 with ref 48 and 49.

4. line 44. FFAs?

5. Indicate the alleles responsible for susceptibility when specific SNPs are cited throughout the text, as is done on the page. 6; line 26.

ej, p. 12 line 26, 39.

6. p. 6 line 38.  reference number is missing. Page 8, Line 45

7. Specify in the text US as ultrasound the first time it is cited.

8. The manuscript would benefit from illustrations and summary tables with essential concepts and ideas.

Author Response

The authors review the factors that affect survival in patients with HCC. Some minor issues could be improved:

We thank the Reviewer for her/his insightful revision of the manuscript and for the suggestions.

Please see below the point-to-point answers to the Reviewers. Changes in the manuscript are marked up using the “Track Changes” (which are in red) and highlighted in yellow.

Point 1: Could the authors refer to how the Hispanic ethnic group is more potentiated in carcinogenesis by HCV?

Response 1: We thank the Reviewer for this interesting comment. We added a possible explanation of how ethnicity affects HCC carcinogenesis in patients with HCV.

Point 2: Some references on HCV would be interesting to consider in this review:

DOI: 10.1016/j.cgh.2022.06.032

doi: 10.1016/S1470-2045(22)00078-X.

https://doi.org/10.1111/acer.13013

DOI : 10.6002/ect.2017.0302

doi: 10.5114/aoms.2018.80651

https://doi.org/10.3171/2022.8.FOCUS2255

Response 2: We thank the Reviewer for having suggested these contributions. Our review focused primarily on HCC development across aetiologies and surveillance, especially in NAFLD-related HCC. Nevertheless, in accordance with the Reviewer’s suggestion, we have now improved the revised version of the manuscript by adding a comment about liver transplantation in alcoholic cirrhotic patients.

Point 3: Authors should indicate the reference number throughout the document after the citation to facilitate reading and searching. Page, 4 et al., line 4 and 6, line 31 and 34 with ref 48 and 49.

Response 3: We apologize with the Reviewer for the imprecisions. As suggested, we have now added the reference numbers after the citations.

Point 4: line 44. FFAs?

Response 4: We thank the Reviewer for this observation. We first used the FFAs abbreviation instead of free fatty acids in page 2, line 10 when we put the abbreviation in parentheses.

Point 5: Indicate the alleles responsible for susceptibility when specific SNPs are cited throughout the text, as is done on the page. 6; line 26. ej, p. 12 line 26, 39.

Response 5: We really thank the Reviewer for this comment. We indicated the at-risk alleles for all genetic variants cited throughout the manuscript.

Point 6: p. 6 line 38.  reference number is missing.

Response 6: We apologize for the inaccuracy. We have now corrected the reference number in the revised manuscript.

Point 7: Page 8, Line 45. Specify in the text US as ultrasound the first time it is cited.

Response 7: We thank the Reviewer for her/his comment. On page 2, line 26, we first abbreviated ultrasound as US.

Point 8: The manuscript would benefit from illustrations and summary tables with essential concepts and ideas.

Response 8: We completely agree with the Reviewer. We have now ameliorated the draft of the manuscript by adding four new tables (Table 1, 2, 3 and 4) and a figure (Figure 2), to better summarize the focus of this review.

Reviewer 3 Report

The authors in their Review paper aimed to explore the differences in HCC features between NAFLD and non-NAFLD patients and those between NAFLD-HCC developed in cirrhotic and non-cirrhotic liver.

The manuscript sound more like a book chapter as a review paper. More updated data in form of Algorithms, table and figures should be absolutely added before further consideration. Only one poor figure is not sufficient.

Major points:

1) It would be essential to add a Table summarizing  the main and updated genetic mechanisms in non-alcoholic fatty liver disease (NAFLD)-related HCC pathogenesis.

2) Could the authors show an algorithm for stratifying the risk of hepatocellular carcinoma and surveillance in patients with and without NAFLD.

3) What about  randomized controlled trials on systemic therapy for advanced hepatocellular carcinoma. Please describe them in text and summarize in table form.

4) As well known, the growing body of evidence concerning the prevention of HCC by lifestyle habits and behaviors. Please focus more on this issue in the text and their role in incidence and outcomes on HCC (related and not related to NAFLD). Here also for the reader data is preferred to be summarized in a table form

Minor points:

Figure 1: It should contain more data information (recent and updated data). Did the authors get this figure from other article ? if yes provide the reference, if no please indicate the program used to design it .

Author Response

The authors in their Review paper aimed to explore the differences in HCC features between NAFLD and non-NAFLD patients and those between NAFLD-HCC developed in cirrhotic and non-cirrhotic liver. The manuscript sound more like a book chapter as a review paper. More updated data in form of Algorithms, table and figures should be absolutely added before further consideration. Only one poor figure is not sufficient.

We thank the Reviewer for the comment. We have now implemented the revised version of the manuscript by adding Table 1. ‘Differences in HCC prevalence according to predisposing background, risk factors and preventing strategies across the diverse aetiologies’, Table 2 ‘Schematic list of the main inherited variations related to NAFLD-HCC’, Table 3 ‘Similarities and differences in HCC carcinogenesis between cirrhotic and non-cirrhotic NAFLD patients’ and Table 4 ‘Non-invasive tests (NITs) for stratifying NAFLD patients according to the risk of developing HCC’. Moreover, as suggested by Reviewer 1, we have now added a new paragraph ‘5.4. Novel strategies to non-invasively assess NAFLD-HCC: proteins and receptors’ and Figure 2Current and future diagnostic tools for the diagnosis of HCC in non-cirrhotic NAFLD patients’.

Major points:

  • It would be essential to add a Table summarizing the main and updated genetic mechanisms in non-alcoholic fatty liver disease (NAFLD)-related HCC pathogenesis.

We thank the Reviewer for the suggestion. We have now added Table 2 in which we have summarized the most known genetic risk factors associated with NAFLD-HCC.  

  • Could the authors show an algorithm for stratifying the risk of hepatocellular carcinoma and surveillance in patients with and without NAFLD.

We have now expanded the paragraph ‘6. New proposed scores to estimate the risk of NAFLD-HCC ‘ by explaining other scores, currently under evaluation, to stratify NAFLD patients according to the risk of HCC onset and by citing more updated contributions in this field. In addition, we have also summarized the main applicable scoring systems in Table 4. Non-invasive tests (NITs) for stratifying NAFLD patients according to the risk of developing HCC.

  • What about randomized controlled trials on systemic therapy for advanced hepatocellular carcinoma. Please describe them in text and summarize in table form.

We are aware that the topic of novel systemic therapies for advanced HCC is, for sure, deemed of interest. However, in our opinion, we believe that an additional paragraph in this review may divert from the main focus of the manuscript. In addition, we have recently published a review on this topic (doi: 10.3390/ijms23052707).

  • As well known, the growing body of evidence concerning the prevention of HCC by lifestyle habits and behaviors. Please focus more on this issue in the text and their role in incidence and outcomes on HCC (related and not related to NAFLD). Here also for the reader data is preferred to be summarized in a table form.

We really thank the Reviewer for the suggestion. We have now added a paragraph concerning the preventing factors of liver carcinogenesis across the etiologies. Moreover, we added these concepts also in Table 3.

Minor points:

Figure 1: It should contain more data information (recent and updated data). Did the authors get this figure from other article? if yes provide the reference, if no please indicate the program used to design it 

We have now updated the manuscript according to the Reviewer’ suggestions. All figures are originals, and they are not reached by other articles. The Figures are designed in ppt and then they are inserted in the manuscript.

Round 2

Reviewer 1 Report

The authors improved the manuscript as requested

Author Response

The authors improved the manuscript as requested.

We thank the Reviewer for his/her positive comment about the manuscript.  

Reviewer 3 Report

The revised version of this review is improved.
Minor points:
Figure 1 and 2: First, the resolution of both figure is yet not improved. Second, why both figures were not combined in one Figure divided in 2 panels (risk factor and diagnostic tools)? Third arrows should be defined in the abbreviations under the figure

Author Response

Figure 1 and 2: First, the resolution of both figure is yet not improved. Second, why both figures were not combined in one Figure divided in 2 panels (risk factor and diagnostic tools)? Third arrows should be defined in the abbreviations under the figure

We thank the Reviewer for his/her positive comments about the manuscript.  We have now combined Figure 1 and 2 together and defined the arrows under the figure.